# Integrative Computational Modeling of Cardiomyocyte Calcium Handling and Cardiac Arrhythmias: Current Status and Future Challenges

**DOI:** 10.3390/cells11071090

**Published:** 2022-03-24

**Authors:** Henry Sutanto, Jordi Heijman

**Affiliations:** 1Department of Cardiology, CARIM School for Cardiovascular Diseases, Maastricht University, 6229 ER Maastricht, The Netherlands; jordi.heijman@maastrichtuniversity.nl; 2Department of Physiology and Pharmacology, SUNY Downstate Health Sciences University, Brooklyn, NY 11203, USA

**Keywords:** computational modeling, cardiomyocyte, calcium handling, cardiac arrhythmia, electrophysiology, integrative experiment, cardiovascular

## Abstract

Cardiomyocyte calcium-handling is the key mediator of cardiac excitation-contraction coupling. In the healthy heart, calcium controls both electrical impulse propagation and myofilament cross-bridge cycling, providing synchronous and adequate contraction of cardiac muscles. However, calcium-handling abnormalities are increasingly implicated as a cause of cardiac arrhythmias. Due to the complex, dynamic and localized interactions between calcium and other molecules within a cardiomyocyte, it remains experimentally challenging to study the exact contributions of calcium-handling abnormalities to arrhythmogenesis. Therefore, multiscale computational modeling is increasingly being used together with laboratory experiments to unravel the exact mechanisms of calcium-mediated arrhythmogenesis. This article describes various examples of how integrative computational modeling makes it possible to unravel the arrhythmogenic consequences of alterations to cardiac calcium handling at subcellular, cellular and tissue levels, and discusses the future challenges on the integration and interpretation of such computational data.

## 1. Cardiomyocyte Calcium Handling: The Complex Hub of Excitation and Contraction

Calcium plays a major role in both excitation and contraction of atrial and ventricular cardiomyocytes. The influx of calcium via L-type calcium channel (LTCC) mediates a much larger calcium release from the sarcoplasmic reticulum (SR) to the cytoplasm through calcium-induced calcium release (CICR), which initiates myriads of processes within a cardiomyocyte, including the activation of calcium-dependent signaling molecules (e.g., calmodulin, calcineurin, calpain, etc.), transmembrane ion channels, calcium-handling proteins and contractile machineries of the cardiomyocyte. During diastole, calcium ions are partly stored back in the SR and partly extruded from the cell via the sodium-calcium exchanger (NCX). Under pathological conditions, these complex processes can be disrupted, creating substrates and triggers for cardiac arrhythmias [1]. There are three major calcium-dependent arrhythmogenic mechanisms: initiation of afterdepolarizations (i.e., early afterdepolarizations [EADs] and delayed afterdepolarizations [DADs] leading to triggered activity [TA]), direct and indirect ion-channel modulation, and the promotion of structural remodeling [1]. Those pathological processes can modify action potential (AP) properties (e.g., action potential duration [APD], resting membrane potential [RMP] and effective refractory period [ERP]) and cell-to-cell coupling, further altering tissue-level conduction velocity. In turn, these electrophysiological alterations may promote ectopic activity, reentrant waves and cardiac arrhythmias (Figure 1). In addition, in sinoatrial (SA) node cells, calcium handling strongly modulates pacemaking activity through a calcium clock involving spontaneous SR calcium leak, which in turn activates NCX to promote diastolic depolarization and automaticity. Accordingly, calcium-handling abnormalities may also promote sinus node dysfunction [2,3,4].

Due to the complex interacting and tightly-controlled calcium-mediated processes within the heart, it is experimentally challenging to study the exact role of calcium in arrhythmogenesis. There are several conflicting evidences with regard to the roles of calcium and calcium-handling proteins in cardiac pathologies. For example, sarcoendoplasmic reticulum calcium ATPase (SERCA) upregulation is considered proarrhythmic by increasing calcium leak and spontaneous calcium release events (SCaEs) due to store overload-induced calcium release (SOICR) [5]. However, at the same time, evidence suggests that SERCA stimulation may reduce the likelihood of SCaEs and triggered activity by attenuating the communication between ryanodine receptor (RyR2) clusters, elevating the intra-SR threshold for the generation of calcium waves and slowing calcium-wave propagation [6]. This antiarrhythmic behavior of SERCA stimulation was documented in several studies using heart failure (HF) rats [7] and ischemia-reperfusion porcine models [8]. Likewise, in atrial cardiomyocytes from patients with long-standing persistent atrial fibrillation (AF), some studies reported RyR2 dysfunction and increased SCaE incidence [9,10,11,12,13], while others showed reduced SCaE incidence and calcium-signaling silencing (Figure 2) [14,15,16,17]. These two examples highlight the complexity of calcium-mediated arrhythmogenic processes within a cardiomyocyte, with numerous feedforward and feedback mechanisms operating over different temporal and spatial scales, which laboratory experiments have not been able to fully resolve.

## 2. Integrative Experimentation: Is It Necessary?

There are at least three levels at which laboratory experiments in cardiovascular research can be performed: in vitro, ex vivo (e.g., the whole-heart Langendorff setup) and in vivo. Although these experiments have been instrumental in investigating different aspects of calcium-mediated arrhythmogenesis, each of them has their own limitations. In vitro experiments are widely used to study the arrhythmogenic consequences of calcium-handling abnormalities at the subcellular and cellular levels. Due to the limited availability of primary human cardiomyocytes, cardiomyocytes derived from various animal models, human induced pluripotent stem cells and, more recently, immortalized human cardiomyocytes, have been used [18,19]. However, these systems are not identical to primary human cardiomyocytes, and may present potentially different intracellular calcium handling and calcium-dependent effects. Moreover, the absence of complete and mature signaling pathways within some of these cells may also affect the findings. However, even when using primary (adult) human cardiomyocytes, several limitations have to be considered. The availability of non-diseased donor hearts is limited, so, samples from patients undergoing cardiac surgery who have an extensive history of cardiovascular disease, are commonly used. In addition, isolated cardiomyocytes lack regulation by systemic modulators, such as the autonomic nervous and humoral systems, which have been shown to hold an important role in calcium-mediated arrhythmogenesis. Furthermore, single-cell experiments also lack intercellular coupling, which might modify the observed cellular effects. The role of electrotonic coupling has previously been studied in canine wedge preparations, in which cells close to the site of pacing experienced the greatest electrotonic load and therefore had the maximum attenuation of AP upstroke amplitude. When electrotonic load due to propagation was reduced, upstroke amplitude was markedly enhanced [20]. Likewise, the inhibition of transient-outward potassium current (I_to_) significantly reduced calcium-transient amplitude in single-cell simulations, while the reduction of calcium-transient amplitude following I_to_ blockade was minimal in tissue simulations due to electrotonic load [21].

Although ex vivo experiments could potentially solve some of the aforementioned limitations of in vitro experiments, several challenges remain. Since the examined heart has to be detached from the body, it has to be denervated and therefore might lose some of the autonomic nervous control. Moreover, this type of experiment does not permit the study the underlying cause of the calcium-dependent arrhythmia, which are commonly found at more microscopic scales, at the exact timescale when the arrhythmia occurs or in the same cells. Simultaneous optical mapping can be performed in Langendorff-perfused hearts to capture the membrane potential and the intracellular free calcium at the same time [22,23,24]. However, at the moment, this approach is limited to these two parameters (membrane potential and bulk calcium transient), has limited spatial and temporal resolution, and is unable to image the depth of the cardiac walls. Therefore, it is often unable to unravel the causality of calcium-dependent cardiac arrhythmias.

Moreover, it is currently challenging to non-invasively visualize calcium in vivo. Some efforts using multiphoton microscopy technique [25] and optical mapping [26] have enabled recordings of membrane potential and calcium in an autonomically intact heart. However, they required surgical access to the heart, hindering the application of the techniques in humans. In the future, noninvasive photoacoustic imaging [27] might be useful to visualize and quantify the electrophysiological determinants at molecular and cellular levels [28]. Despite the ability of in vivo experiments to show the contribution of autonomic nervous system to various calcium-dependent processes, several limitations remain, including ethical considerations, cost and preparation time. These limitations typically preclude the use of large numbers of samples, making results more susceptible to random fluctuations. Therefore, currently available non-integrative laboratory experiments are not adequate to fully elucidate the mechanisms underlying calcium-mediated arrhythmogenesis and another means of integrative experimentation is required to unravel the exact pathophysiology of calcium-induced arrhythmia.

## 3. Why Do We Need Integrative Computational Modeling?

For more than 50 years, computational modeling of cardiac cellular electrophysiology has been shown to be beneficial in identifying the drivers of cardiac arrhythmias at the cellular scale. By employing the perfect control and observability of in silico models, the key determinants of arrhythmogenesis can be identified and managed. Computational modeling is also cost-effective, and with the advancement of computational power and the development of in silico tools such as Chaste [29], OpenCARP [30] and Myokit [31], models at the subcellular to organ levels can be simulated in a fairly short timeframe. For example, we previously demonstrated some of the benefits of computational modeling to better understand the pathophysiology of calcium-dependent arrhythmia. Employing our state-of-the-art spatial calcium-handling model, we showed that heterogeneity of the distributions of calcium-handling proteins has a significant impact on the propensity of SCaEs [32]. Moreover, using our model, we were able to manipulate the arrangement of the axial tubules, attach and detach the experimentally observed concomitant RyR2 hyperphosphorylation, and selectively modify the subcellular distributions of these calcium-handling components, which at present cannot be carried out experimentally. With this approach, we could show, for the first time, the consequences of different locations and number of axial tubules for atrial calcium wave propagation, the magnitude of the contribution of RyR2 hyperphosphorylation, and the role of lateral RyR2 bands and inter-band RyR2 clusters in calcium wave propagation and how they may contribute to the susceptibility to SCaEs and DADs [32]. We also employed this spatial calcium-handling model to investigate the consequences of both calcium-handling abnormalities and acute transient inflammation on the propensity of SCaEs in the setting of post-operative AF (POAF). Our findings confirmed that inflammation is a prerequisite to trigger cellular proarrhythmicity in the vulnerable preexisting substrate of POAF patients [33].

Numerous in silico cardiomyocyte models for a wide range of species have been developed [34] and used to evaluate the pro- and antiarrhythmic effects of pharmacological interventions [35,36,37]. Using our novel tool (Maastricht Antiarrhythmic Drug Evaluator; MANTA [38]), we showed that interspecies differences in ion-channel function may significantly affect the cellular response to antiarrhythmic drugs (AADs). Furthermore, our tool may also provide a hint on potential drug-induced proarrhythmia exhibited by various AADs under specific pathophysiological conditions [38]. Similarly, using in silico models, we also demonstrated the difference between canine and human ventricular cardiomyocytes in facilitating EAD generation in the presence of reduced repolarization reserve [39]. Moreover, in a population of 1000 models, we also revealed that β-adrenergic stimulation may limit the proarrhythmic behavior of APD-prolonging drugs by restoring the repolarization reserve, although in the presence of increased I_CaL_ window, β-adrenergic stimulation could be detrimental, further highlighting the importance of considering dynamic modulation of cardiac electrophysiology in cardiomyocyte models [39].

Despite the benefits of subcellular or cellular in silico models, the clinical applicability of such findings is limited. Although we showed an increase in SCaEs at the (sub)cellular level [32], their effects at higher levels, such as tissue and organ levels, are less well-known. Likewise, although MANTA [38] was able to demonstrate potential drug-induced arrhythmia at the cellular level, the effect might be eliminated by strong intercellular electrotonic coupling or cell-to-cell variability at higher scales (Figure 3). Therefore, the integration of in silico models into a multiscale modeling approach is needed [40]. We previously employed integrative modeling to study the potential pro- and antiarrhythmic effects of alcohol consumption [41]. Ethanol is known to alter multiple cardiac ion channels, calcium-handling proteins and gap-junction coupling. Using a multiscale model, we could show that ethanol-induced remodeling of I_K1_ contributed to the protective effects produced by low ethanol concentrations by slightly prolonging atrial APD and lowering the total arrhythmogenic risk at the tissue level. We also showed the consequences of disease-associated electrical and structural remodeling on the ethanol-induced arrhythmogenesis. Finally, we showed that cell-to-cell variability in ethanol-induced I_K1_ remodeling was an important determinant for the tissue phenotype. Therefore, characterization of such variability is needed to accurately predict the effect of ethanol in cardiac electrophysiology [41].

## 4. All Models Are Wrong, but Some Are Useful

George E.P. Box, a British statistician, once said “*all models are wrong, but some are useful*”, which indicates that no in silico model is perfect and that each model has its own limitations and uncertainties. Indeed, models are by definition a simplification of reality. They can become irrelevant over time, e.g., due to the discoveries of new experimental findings, simultaneously allowing newer and more advanced models to develop. However, cardiac electrophysiology is a quantitative science based on well-established physical principles and with large amounts of experimental data. Existing cardiomyocyte models have been constructed based on the available experimental data at the moment when the models were developed and they have been useful to address the specific research questions that they were intended to answer. Moreover, the most interesting lessons are learned when a model is discovered to be incorrect, as this reflects a gap in knowledge or error in our underlying assumptions. As such, one could consider that the statement “*All models are imperfect, but they are nonetheless useful*” would be more appropriate. Nevertheless, several areas for improvement remain. Here, we describe four common limitations of in silico models, which require additional attention in future research: temperature dependence, interspecies diversity, intra-/inter-individual heterogeneities and model-dependent effects.

### 4.1. Temperature Dependence

It is essential for in vitro experiments to be conducted at body temperature (approximately 34–37 °C) to produce results closest to physiological conditions. However, sometimes it is challenging due to the increased cell instability at higher temperatures, which forces the experiments to be carried out at lower temperature (e.g., room temperature). Temperature differences alter the function of several calcium-handling proteins. For example, an increase in temperature from 24 to 37 °C increased the rate-constant of decay of the calcium-transient (CaT) and caffeine-induced calcium-transient (cCaT) in both rat and guinea-pig cardiomyocytes, reflecting altered NCX and SERCA function at higher temperature [42]. Similarly, temperature changes have been reported to affect the properties of calcium sparks in rat cardiomyocytes. A reduction of experimental temperature from 35 to 10 °C increased the frequency of calcium sparks and reduced their amplitude, while prolonging the time-to-peak and decay of calcium sparks [43]. Such findings emphasize the need to carefully observe the data prior to the incorporation into in silico models.

To accommodate thermodynamic effects, several computational models have incorporated scaling factors, typically based on a Q10 factor (the change in rate for a 10-degree increase in temperature), into the ionic-current equations. However, the data to validate these factors are scarce and thus may affect the robustness of computational studies when simulations are performed at temperatures other than those at which the data were obtained. Moreover, the temperature-dependent effects may be distinct for different model components and experimental data from different sources may be obtained at different temperatures, making their integration in computational models challenging. Of note, similar considerations may hold for other experimental variables, such as the composition of the intracellular and extracellular solutions [44].

### 4.2. Interspecies Diversity

The availability of human cardiomyocyte samples is often limited. Therefore, experimental cardiac cellular electrophysiological research may benefit from the use of heterologous expression system (e.g., human embryonic kidney cells [HEK293], Chinese hamster ovary [CHO], *Xenopus laevis* oocytes, etc.), in which a specific ion-channel of interest can be analyzed using patch-clamp experiments. Alternatively, cardiomyocytes can be obtained from various animal models, including mouse, rat, rabbit, guinea-pig, pig, and dog. However, each species has a different composition of cardiac ion channels that may affect the observations and experimental results (Figure 4) [45,46]. For example, mice express little rapid delayed-rectifier potassium current (I_Kr_) and slow delayed-rectifier potassium current (I_Ks_), which limits their use for drug-induced proarrhythmia research [45,47]. We previously demonstrated such interspecies differences in the AP response to AADs using MANTA [38]. Following the application of dofetilide (a class III AAD, primarily blocking I_Kr_), canine left ventricular (LV) and human LV models revealed the biggest APD prolongation, while the mouse LV model showed no change in AP properties. Similarly, the calcium-channel blocker verapamil (a class IV AAD), which also blocks I_Kr_, displayed species-specific responses in the models. In mouse models, the drug slightly prolonged the APD, while in the guinea-pig and rabbit, it shortened APD. Interestingly, in the canine model, the drug had opposite effects at low and high concentrations, with low concentrations prolonging APD and high concentrations shortening APD [38]. Thus, understanding the characteristics (i.e., ion-channel compositions) of the species/models of interest is essential.

Unfortunately, the availability of experimental data from humans (or even from the same animal model) for computational analysis are often limited. Therefore, most in silico models are based on experimental data from multiple animal models, which could potentially influence the behavior of in silico models [53,54]. In these models, we often assume that the observed effects are conserved across species. Although in some cases this assumption is valid, prudent interpretation of the results is necessary.

### 4.3. Intra- and Inter-Individual Heterogeneities

The pathogenesis of cardiac arrhythmias has not been fully elucidated due to the complex interaction between signaling molecules, calcium-handling proteins, ion channels and other proarrhythmic substrates within the heart. In particular, there are considerable intra- and inter-individual heterogeneities in cardiac electrophysiological properties, which influence the effectiveness and safety of currently available one-size-fits-most therapeutic strategies for cardiac arrhythmias [55]. Inter-individual variability is evident even among healthy individuals. However, the clinically observed magnitude of such inter-individual heterogeneities might be overestimated due to the considerable overlap with unexplored intra-individual heterogeneity. Intra-individual variation can be the result of distinct interactions between electrical impulse determinants at the cellular, tissue (e.g., gap-junction coupling and anisotropy) and organ levels (e.g., fiber arrangement, wall thickness, structural remodeling and autonomic innervation). At the molecular level, the spatial heterogeneity of the distribution and expression of ion channels and signaling molecules, as well as the interactions between two or more tightly co-expressed genes and the integration of multiple proteins in macromolecular complexes may contribute to the functional variability observed at the cellular and tissue levels [56]. In the healthy heart, the effect of single-cell variability may be muted by strong electrotonic coupling [57], while in the diseased heart with altered intercellular coupling, the consequences of such variability are largely unknown and yet to be investigated. Comprehensive characterizations of this molecular and cellular heterogeneity have so far not been performed.

Computationally, inter-/intra-individual heterogeneity is (partly) addressed by employing a population of models, incorporating variability of specific parameters of interest (typically the maximum conductance of ionic currents, reflecting differences in the expression level of ion channels) [58]. Using this population-based approach, the inter-individual variability of ionic currents can be accommodated and more representative results can be obtained. However, the exact magnitude of such variability is unknown and therefore relies on predefined assumptions. In our previous study [41], in addition to the population modeling approach to confirm our cellular findings, we also performed multiscale in silico simulations to study the role of cell-to-cell variability at the tissue level. We aimed to study whether the ethanol-associated increase in reentrant arrhythmia vulnerability is affected by the variability of ethanol-induced I_K1_ remodeling. Our simulations revealed that the proportion of cells incorporating an ethanol-induced increase or decrease of I_K1_ in the virtual tissue modulated the behavior of the reentrant arrhythmias [41]. This finding highlights the need for considering cell-to-cell variability at the higher scale (e.g., tissue or organ level) models.

### 4.4. Model-Dependent Effects

Another aspect that complicates the direct interpretation of in silico data is the presence of model-dependent effects. As previously demonstrated [59], there were considerable differences in the behavior of in silico models of atrial cardiomyocytes in response to ionic perturbations. Figure 5 exemplifies these inter-model differences in AP and CaT properties in canine and human ventricular cardiomyocyte models. Such differences could be due to several contributing factors, including the intercellular heterogeneities and distinct ionic formulation in those in silico models.

Each computational model is commonly validated to data from a subset of samples (can be animals or humans) collected during in vitro, ex vivo or in vivo experiments. Due to the limited number of samples available or heterogenous sample size between studies, it could be that the presented data does not cover the complete intercellular heterogeneity. As a consequence, there could be a notable difference in the validated parameter values between models, which may impact the multiscale behavior of the in silico models. This issue can be partly resolved by applying population modeling approach, as discussed in the previous subsection. Another factor that may be involved in the model-dependent effects is the distinct ionic formulation between models. Although the output of each formulation is commonly validated to experimental data, to what extent and under which conditions those formulations should be validated to ensure the consistency between models remain unknown. This uncertainty results in the potentially variable behavior across in silico models under certain (environmental) conditions or perturbations, which could also impair the accuracy of model predictions.

## 5. The Future Outlook of Integrative In Silico Modeling

In some cases, 2-dimensional tissue simulations may be adequate to address the arrhythmogenic consequences of calcium handling abnormalities. By employing such tissue simulations, the significance of cell-to-cell interaction can be elucidated with a fairly reasonable computational cost [67]. However, clinical arrhythmias are organ-level phenomena and 2-dimensional results may not be representative of the actual condition in the organ level due to the absence of other supporting components, such as extracellular matrices, Purkinje fibers, muscle-fiber orientation and spatial gradients in electrophysiological properties (e.g., between left and right, or between endocardial, mid-myocardial and epicardial layers) [68]. Those organ-level components might also play an important role in wave propagation within the heart [69]. Therefore, organ-level modeling incorporating detailed structures of cardiac electrophysiology may ultimately be needed for studying cardiac arrhythmias.

However, despite its potential benefit for arrhythmia research, organ-level modeling is computationally intense, often requiring supercomputing clusters to operate, which hinders its broad application [68]. To overcome this issue, (sub)cellular models have had to be significantly simplified. As an illustration, on a normal personal computer, a simulation using the detailed spatial subcellular calcium-handling model with stochastic ion channel gating could take more than 12 h to complete, while a cellular simulation using a deterministic common-pool AP model would take less than two minutes to compute. Meanwhile, a simulation of 400 × 400 cells with detailed ionic currents could last for more than 8 h. This computational burden limits the use of such detailed models at tissue- and organ-level scales. Simpler alternatives have been proposed at the cost of losing realism in the mathematical description, e.g., using phenomenological models, which reduce the number of state-variables, substituting the actual ionic current descriptions by simple mathematical equations [68]. However, such simplifications may limit the ability of the organ-level model to mimic the precise conditions observed experimentally, as well as the mechanistic interpretation of the results, particularly in the presence of genetic mutations altering the gating components of specific cardiac ion channels or calcium-handling proteins. Therefore, until now, the application of organ-level modeling in pathologies requiring complex biophysical details has been restricted to a limited number of labs worldwide. Nonetheless, several approaches have been proposed to develop simplified phenomenological models capturing key properties of spatial calcium-handling models [70]. In addition, computing power continues to advance and organ-level modeling will likely become feasible to investigate the arrhythmogenic consequences of calcium-handling abnormalities.

Despite the aforementioned challenges of current integrative computational modeling approaches, there is a growing interest in more personalized modeling approaches that accommodate inter-individual heterogeneities to unravel patient-specific mechanisms and develop tailored therapy of cardiac arrhythmias. Such comprehensive personalized approaches have been partially performed, for example to guide catheter ablation in AF [71] and ventricular fibrillation [72] by incorporating cardiac imaging-derived patient-specific fibrotic regions into organ-level models and simulating the effects of ablation lesions, which were then confirmed during a clinical electrophysiological study. In the future, additional personalization based on genetic, cellular (e.g., induced pluripotent stem-cell-derived cardiomyocytes, biomarkers and electrolytes) and functional (e.g., electrocardiogram [ECG] and electrophysiological mapping) data (Figure 6), as well as integration with emerging machine-learning approaches based on the large amounts of cardiovascular data that are being generated, may help to realize the vision of the ‘digital twin’ that can be used to improve patient care [73]. Validation of model-based care in clinical trials and instructions on appropriate use will be essential for clinical adoption. However, to realize widespread use of such mechanistic, data-driven and hybrid models, attention should also be paid to user-friendliness, integration in routine clinical workflows and interpretability of results.

## 6. Conclusions

Calcium-handling abnormalities have multiscale consequences that may promote cardiac arrhythmias. Computational modeling is a powerful tool to investigate the arrhythmogenic mechanisms and consequences of alterations in cardiac calcium handling. Here, we discussed how integrative computational modeling of calcium handling can provide new insights into arrhythmia mechanisms, although numerous challenges remain. Future arrhythmia research will benefit from organ-level modeling of calcium-handling abnormalities, and, more importantly, an integrative patient-specific modeling approach. Such advances will certainly improve the accuracy and predictability of computational models and increase their clinical relevance. John von Neumann stated that “*With four parameters I can fit an elephant, and with five I can make him wiggle his trunk*”. In the end, one extra sentence deserves to be added to this statement: “*… but even with thousands of parameters, I cannot make him alive.*”, indicating that in silico models will always be in silico and therefore, synergy between computational modeling, biological experiments and clinical research remains an absolute prerequisite.

## Figures and Tables

**Figure 1 cells-11-01090-f001:**
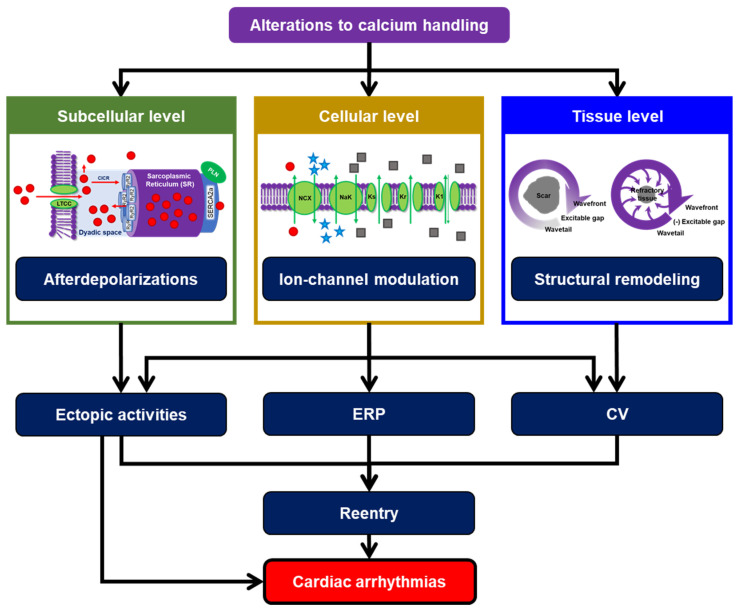
Alterations to calcium handling as a key determinant of cardiac arrhythmias. Subcellular calcium-handling abnormalities may initiate delayed afterdepolarizations (DADs) and triggered activity. At the cellular level, modifications of calcium-handling proteins can directly or indirectly affect cardiac ion-channel function, leading to substantial changes in AP properties. At higher spatial scales, calcium-handling abnormalities may promote structural remodeling, allowing the generation of reentrant waves. Together, those processes may create vulnerable substrates for cardiac arrhythmias. (CICR = calcium-induced calcium release; CV = conduction velocity; ERP = effective refractory period; K1 = inward-rectifier potassium channel; Kr = rapid delayed-rectifier potassium channel; Ks = slow delayed-rectifier potassium channel; LTCC = L-type calcium channel; NCX = sodium-calcium exchanger; NKA = sodium-potassium adenosine triphosphatase; PLN = phospholamban; RyR2 = ryanodine receptor type 2; SERCA2a = sarco/endoplasmic reticulum calcium adenosine triphosphatase-2a).

**Figure 2 cells-11-01090-f002:**
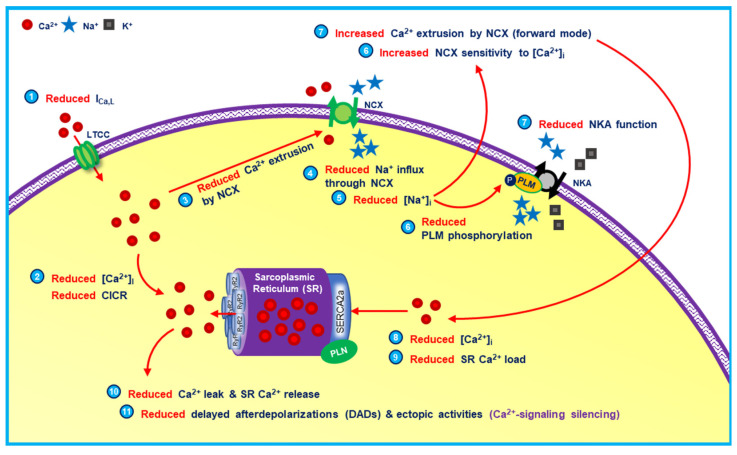
The proposed mechanisms involved in low intracellular sodium-induced calcium silencing. The reduction of LTCC function following rapid atrial pacing/persistent AF has been proposed as the cause of low intracellular sodium, which further induced the calcium extrusion via NCX and lowered the frequency of SR calcium leak and SCaEs, ultimately leading to calcium-signaling stabilization/silencing in the long term. (CICR = calcium-induced calcium release; I_Ca,L_ = L-type calcium current; LTCC = L-type calcium channel; NCX = sodium-calcium exchanger; NKA = sodium-potassium adenosine triphosphatase; PLM = phospholemman; PLN = phospholamban; RyR2 = ryanodine receptor type 2; SERCA2a = sarco/endoplasmic reticulum calcium adenosine triphosphatase-2a).

**Figure 3 cells-11-01090-f003:**
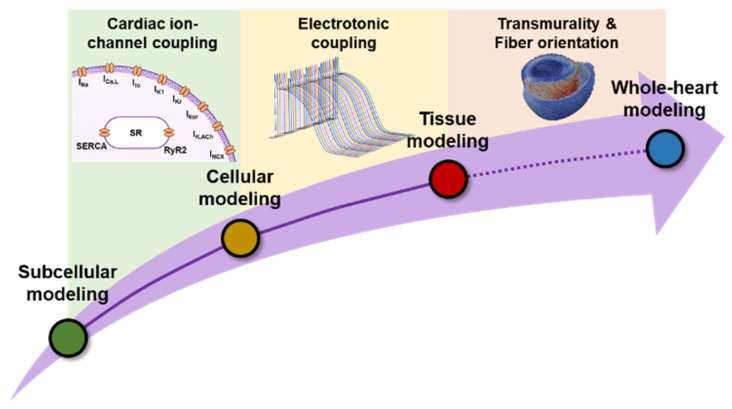
Schematic illustration of multiscale computational modeling. To integrate individual modeling scales into a holistic model, a subcellular model of cardiomyocyte calcium handling is coupled with other cardiac ion channels to form a cellular model. Subsequently, cell models propagate the electrical impulse through gap junctions to form a 1-dimensional strand or 2-dimensional tissue model. At the organ level, different types of tissues, representing different regions and layers of the heart (e.g., epicardium, mid-myocardium, endocardium, Purkinje fibers, etc.) are combined with fiber orientation and structural remodeling.

**Figure 4 cells-11-01090-f004:**
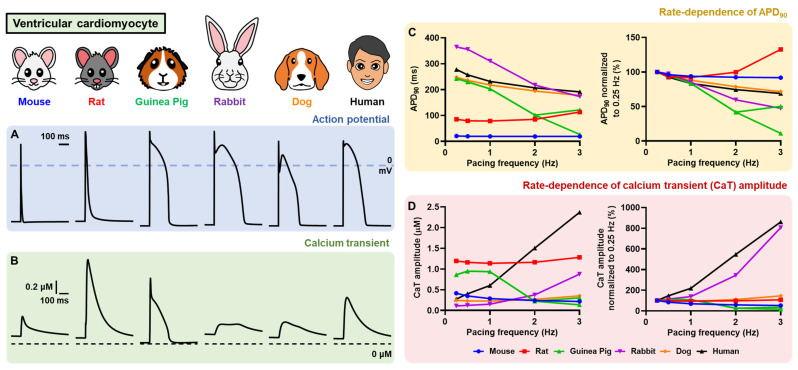
Interspecies difference of AP and calcium transient (CaT) in several computational models of ventricular cardiomyocytes. All simulations were carried out in Myokit [31]. (**A**,**B**) The AP and CaT were stimulated using the default setting of each model at a 1 Hz pacing frequency (basic cycle length [BCL] of 1000 ms) and quasi-steady-state was obtained following 1000 beats of pre-pacing. (**C**,**D**) The rate dependence of APD_90_ and CaT. Mouse model = Bondarenko et al. (2004; [48]), Rat model = Niederer et al. (2007; [49]), Guinea-Pig model = Noble et al. (1998; [50]), Rabbit model = Mahajan et al. (2008; [51]), Dog model = Decker et al. (2009; [21]) and Human model = O’Hara et al. (epicardial; 2011; [52]). By default, the Bondarenko and Niederer models were set at room temperature (22–25 °C), whereas the rest of the models were set at body temperature (±37 °C).

**Figure 5 cells-11-01090-f005:**
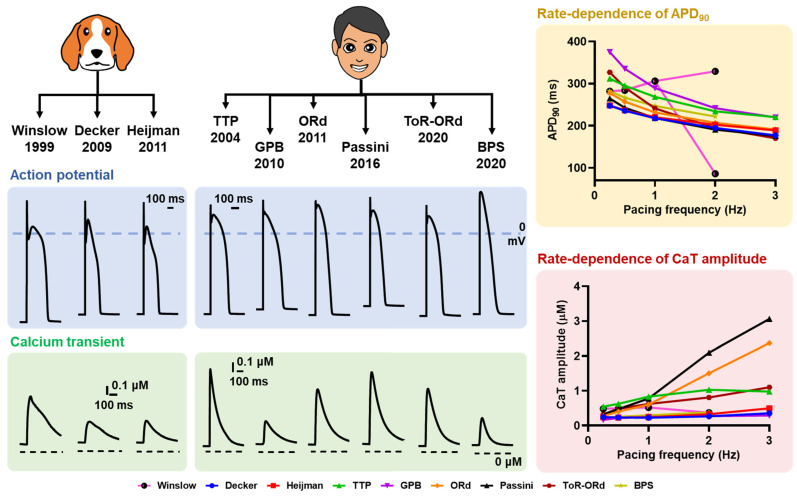
Comparison of APD and calcium transient (CaT) between several canine and human ventricular cardiomyocyte models. The AP and CaT were stimulated using the default setting of each model at a 1 Hz pacing frequency (basic cycle length [BCL] of 1000 ms; the left panels) and quasi-steady-state was obtained following 1000 beats of pre-pacing. (BPS model = [60]; Decker model = [21]; GPB model = [61]; Heijman model = [62]; ORd model = [52]; Passini model = [63]; ToR-ORd model = [64]; TTP model = [65]; Winslow model = [66]).

**Figure 6 cells-11-01090-f006:**
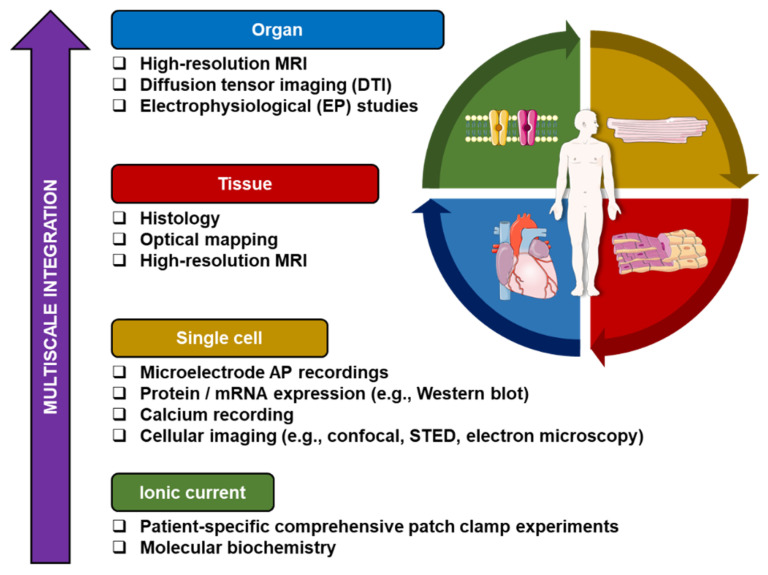
A schematic illustration summarizing multiscale integration of patient-specific data. The “bottom-up” approach starts from the ionic current/molecular level up to the organ level. At each scale, the acquired data can be utilized as an input for computational modeling. Ultimately, a personalized 3-dimensional computational model can be developed and used for patient-specific arrhythmia research. (MRI = Magnetic Resonance Imaging; mRNA = messenger Ribonucleic acid; STED = Stimulated Emission Depletion).

## Data Availability

Not applicable.

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
