# Peer review of "Integrative Computational Modeling of Cardiomyocyte Calcium Handling and Cardiac Arrhythmias: Current Status and Future Challenges"

_cells, 2022, doi:10.3390/cells11071090_

Round 1
Reviewer 1 Report
Using computational modeling, the current study described the role of calcium-handling abnormalities in arrhythmogenesis. Please address the issues listed below.
- Has the author's MANTA been used by others or not? Please provide the reference for assistance.
- Heart complex arrhythmias did not include spontaneous packing. Why?
- Machine-aided learning, including in silico models, has recently become popular. However, it has been challenged by researchers due to limitations in machine operation. For example, MANTA is easily applied by others without author guidance? To each of your statements, "All models are imperfect, but they are still useful," this bias must be addressed first.
- The population modeling approach can be used to back up the proposed viewpoint. Clear indications in computational operation methods are essential. However, it is easily terminated by ambiguous software and tools. Please take this into account.
- Computational modeling is a powerful tool used by clinical cardiologists, and it seems the focus of this report.
- References to your methods may help to strengthen this report.
Author Response
Using computational modeling, the current study described the role of calcium-handling abnormalities in arrhythmogenesis. Please address the issues listed below.
First, we would like to thank the Reviewer for his/her time and for the very constructive comments, which we believe have significantly improved our manuscript. We will attempt to address those comments point-by-point below.
- Has the author's MANTA been used by others or not? Please provide the reference for assistance.
To address this question, we performed a quick search on Google Scholar and found 12 citations associated with our MANTA paper. However, a brief investigation revealed that those publications mainly discussed the consistency of our previous findings using MANTA with theirs or employed the extensive database of IC50 of antiarrhythmic drugs that we constructed from numerous studies, rather than showing that they used MANTA for their projects. As such, we currently cannot provide any reference to previous publications, as requested. Nonetheless, MANTA has been used as a teaching tool in the Systems Biology program in our University (Maastricht University) and some external colleagues have contacted us about MANTA, suggesting their interest in this tool. It is also important to note that MANTA was developed for non-modelers and we currently focus on the application of MANTA as a teaching tool to better understand the mechanism of action of antiarrhythmic drugs rather than a research tool, which is likely not captured by citations in scientific papers.
- Heart complex arrhythmias did not include spontaneous packing. Why?
Thank you for the question. We assume that you meant “spontaneous pacing”? Adult atrial and ventricular cardiomyocytes do not have spontaneous activity (i.e., are not self-beating), as seen, e.g., in neonatal cardiomyocytes, induced pluripotent stem cells or in pacemaker cells. The mechanisms of spontaneous pacing (automaticity) are complex and have been extensively discussed in several literature reviews, including Morad and Zhang (2017) and Lakatta et al. (2010). Briefly, it involves the concerted action of calcium and membrane / voltage clocks. We have now briefly mentioned that calcium handling plays a major role in pacemaking and that calcium-handling abnormalities can contribute to sinus node dysfunction:
“In addition, in sinoatrial (SA) node cells, calcium handling strongly modulates pace-making activity through a calcium clock involving spontaneous SR calcium leak, which in turn activates NCX to promote diastolic depolarization and automaticity. Accordingly, calcium-handling abnormalities may also promote sinus node dysfunction [2-4].”
Morad, M., & Zhang, X. H. (2017). Mechanisms of spontaneous pacing: sinoatrial nodal cells, neonatal cardiomyocytes, and human stem cell derived cardiomyocytes. Can J Physiol Pharmacol, 95(10), 1100–1107. https://doi.org/10.1139/cjpp-2016-0743
Lakatta, E. G., Maltsev, V. A., & Vinogradova, T. M. (2010). A coupled SYSTEM of intracellular Ca2+ clocks and surface membrane voltage clocks controls the timekeeping mechanism of the heart's pacemaker. Circ Res, 106(4), 659–673. https://doi.org/10.1161/CIRCRESAHA.109.206078
- Machine-aided learning, including in silico models, has recently become popular. However, it has been challenged by researchers due to limitations in machine operation. For example, MANTA is easily applied by others without author guidance? To each of your statements, "All models are imperfect, but they are still useful," this bias must be addressed first.
Indeed, we fully agree with the Reviewer that machine learning becomes very popular lately and we actually have discussed the applications, current limitations and the future of such a modality in a separate review article (Heijman et al. 2021).
Heijman, J., Sutanto, H., Crijns, H., Nattel, S., & Trayanova, N. A. (2021). Computational models of atrial fibrillation: achievements, challenges, and perspectives for improving clinical care. Cardiovas Res, 117(7), 1682–1699. https://doi.org/10.1093/cvr/cvab138
It is important to acknowledge that at the moment, we haven’t incorporated machine learning approaches in MANTA. However, as the demand and applicability of MANTA grow, we expect that at some point, we might need to use machine learning to either complement or even to be embedded inside MANTA to fulfill specific research purposes. For example, advances in protein-structure prediction and molecular dynamics based on machine-learning may make it possible to predict the inhibition of any ion channel by a given compound, which could potentially greatly extent MANTA’s library.
We believe that we have provided a clear guidance on how to use MANTA in the supplement of Sutanto et al. 2019 and the users do not need computational skills to operate the tool as they only need to click available buttons to simulate an action potential. Nevertheless, some computer skills and/or instructions may be needed to install the Myokit software that MANTA is based on.
We have added a statement about the potential of machine-learning approaches, but also the need to make models user-friendly and, as much as possible, understandable, to facilitate their wide-spread use and application:
“In the future, additional personalization based on genetic, cellular (e.g., induced pluripotent stem-cell-derived cardiomyocytes, biomarkers and electrolytes) and function-al (e.g., electrocardiogram [ECG] and electrophysiological mapping) data (Figure 6), as well as integration with emerging machine-learning approaches based on the large amounts of cardiovascular data that are being generated, may help to realize the vision of the ‘digital twin’ that can be used to improve patient care [73]. Validation of model-based care in clinical trials and instructions on appropriate use will be essential for clinical adoption. However, to realize widespread use of such mechanistic, da-ta-driven and hybrid models, attention should also be paid to user-friendliness, integration in routine clinical workflows and interpretability of results.”
- The population modeling approach can be used to back up the proposed viewpoint. Clear indications in computational operation methods are essential. However, it is easily terminated by ambiguous software and tools. Please take this into account.
As indicated in response to the previous comment, we agree with the reviewer that clear definition of computational methods, validation, and instructions on (appropriate) use are important and have tried to emphasize this in the revised version of the manuscript.
- Computational modeling is a powerful tool used by clinical cardiologists, and it seems the focus of this report.
Thank you for highlighting this important remark. The use of computational modeling of cardiac electrophysiology is emerging and some of them have been used to guide the decision-making process in the clinic, for example computational studies done by Natalia Trayanova’s group at John Hopkins University. In their studies, they employed organ-level modeling to guide ablation therapy in the clinic. The details of those studies have been discussed in our previous review (Heijman et al. 2021). In addition, other modeling studies, including ours, have been used to unravel the complex mechanisms of cardiac arrhythmias and to improve the understanding of the cellular mechanisms of antiarrhythmic drugs.
Heijman, J., Sutanto, H., Crijns, H., Nattel, S., & Trayanova, N. A. (2021). Computational models of atrial fibrillation: achievements, challenges, and perspectives for improving clinical care. Cardiovas Res, 117(7), 1682–1699. https://doi.org/10.1093/cvr/cvab138
- References to your methods may help to strengthen this report.
Thank you for the suggestion. We have referenced some of our previous works, but have limited this to avoid extensive self-citation. Instead, we have added several sentences about other modeling studies and references to a few key reviews addressing methodologies and applications of computer modeling in cardiac electrophysiology to section 3 of the revised manuscript.
Reviewer 2 Report
Dr Sutanto present a clear, well-structured, and enjoyable-to-read perspective on capabilities and future potential of integrative computational modelling to elucidate the arrhythmogenic consequences of alterations to cardiac calcium handling at subcellular, cellular and tissue levels. The manuscript is written in very fluent English.
I have only one majorish or not-so-minor comment for the author to consider: Chapter 5.2. In my opinion, this chapter doesn’t really add much. It feels like disconnected add-on to an otherwise coherent story. I do realise that personalised modelling is a very current and hot topic. However, as part of this perspective, this chapter only scratches the surface. I wonder if it would serve better for the purpose of this manuscript to condense a note on personalised modelling to 1-2 sentences at the end of the previous chapter and to cite a couple recent reviews on the topic.
Minor comments regarding the style and content:
- Abstract: “calcium-handling” --> “calcium-handling”
- Figure: I would consider some other colour than red for the “Tissue level” box, so that it does not associate with the red “Cardiac arrhythmias” box. How about blue or orange, for example?
- Lines 58-59: Maybe “attenuating” instead of “impairing”?
- Figure 2: “Increased forward mode of NCX” and “Increased Ca2+ extrusion by NCX” are essentially the same thing. I would remove the former. And, if “forward mode” should be mentioned in the figure, it can be included in the latter one e.g. like this: “Increased Ca2+ extrusion by NCX (forward mode)”.
- Line 112: “…cardiac arrhythmias. Moreover, it is …” I would start a new paragraph from “Moreover, it is …”
- Lines 198 and 378: I guess it’s a matter of taste, but to my ear it sounds weird to use “we“, when there is only one author. I would write “I” instead.
- Chapter 4.2: Is there some specific reason to write “degrees Celsius” instead of “°C” in short?
Author Response
Dr Sutanto present a clear, well-structured, and enjoyable-to-read perspective on capabilities and future potential of integrative computational modelling to elucidate the arrhythmogenic consequences of alterations to cardiac calcium handling at subcellular, cellular and tissue levels. The manuscript is written in very fluent English.
We would like to thank the Reviewer for the constructive comments, suggestions and corrections, which we believe have significantly improved our manuscript. Please see our point-by-point response below.
I have only one majorish or not-so-minor comment for the author to consider: Chapter 5.2. In my opinion, this chapter doesn’t really add much. It feels like disconnected add-on to an otherwise coherent story. I do realise that personalised modelling is a very current and hot topic. However, as part of this perspective, this chapter only scratches the surface. I wonder if it would serve better for the purpose of this manuscript to condense a note on personalised modelling to 1-2 sentences at the end of the previous chapter and to cite a couple recent reviews on the topic.
Thank you for this suggestion. We fully agree with the Reviewer that we only touched the surface of integrative personalized modeling and that this is a major (and extremely important) topic that deserves a dedicated manuscript. Based on the reviewer’s suggestion, we have condensed Chapter 5.2 (while adding a few statements on validation and user-friendliness based on comments from Reviewer 1) and integrated it into the remainder of the revised manuscript. Because of this, heading 5.1 has also been removed, to avoid having a chapter with a single sub-section.
Minor comments regarding the style and content:
- Abstract: “calcium-handling” --> “calcium-handling”
- Figure: I would consider some other colour than red for the “Tissue level” box, so that it does not associate with the red “Cardiac arrhythmias” box. How about blue or orange, for example?
- Lines 58-59: Maybe “attenuating” instead of “impairing”?
- Figure 2: “Increased forward mode of NCX” and “Increased Ca2+ extrusion by NCX” are essentially the same thing. I would remove the former. And, if “forward mode” should be mentioned in the figure, it can be included in the latter one e.g. like this: “Increased Ca2+ extrusion by NCX (forward mode)”.
- Line 112: “…cardiac arrhythmias. Moreover, it is …” I would start a new paragraph from “Moreover, it is …”
- Lines 198 and 378: I guess it’s a matter of taste, but to my ear it sounds weird to use “we“, when there is only one author. I would write “I” instead.
- Chapter 4.2: Is there some specific reason to write “degrees Celsius” instead of “°C” in short?
Thank you for these excellent and very helpful corrections, which have been incorporated in the revised version of the manuscript.